# Effectiveness of Sunn Hemp (*Crotalaria juncea* L.) in Reducing Wireworm Damage in Potatoes

**DOI:** 10.3390/insects16070674

**Published:** 2025-06-27

**Authors:** Lorenzo Furlan, Stefano Bona, Roberto Matteo, Luca Lazzeri, Isadora Benvegnù, Nerio Casadei, Elisabetta Caprai, Ilaria Prizio, Bruno Parisi

**Affiliations:** 1Veneto Agricoltura, 35020 Legnaro, Italy; isadora.benvegnu@gmail.com; 2Department of Agronomy, Food, Natural Resources, Animals and Environment, University of Padova, 35020 Legnaro, Italy; stefano.bona@unipd.it; 3Council for Agricultural Research and Economics (CREA)—Research Centre for Cereal and Industrial Crops (CREA-CI), 40128 Bologna, Italy; roberto.matteo22@outlook.com (R.M.); lucalazzeri8@gmail.com (L.L.); nerio.casadei@crea.gov.it (N.C.); bruno.parisi@crea.gov.it (B.P.); 4NRL for Plant Toxins in Food, Food Chemical Department, Istituto Zooprofilattico Sperimentale della Lombardia e dell’Emilia-Romagna (IZSLER), 40127 Bologna, Italy; elisabetta.caprai@izsler.it (E.C.); ilaria.prizio@izsler.it (I.P.)

**Keywords:** *Agriotes*, *Agriotes sordidus*, *Solanum tuberosum*, pyrrolizidine alkaloids, cover crops

## Abstract

Wireworms are a major threat to potatoes. Agronomic prevention, which is always the first IPM strategy to be implemented, can involve bioactive cover crops planted before a susceptible crop. This work assesses the potential for sunn hemp (*Crotalaria juncea*), a tropical leguminous plant, as a cover crop to reduce wireworm damage risk. Trials were conducted in-the-field in semi-natural conditions, with pots used to introduce a set number of reared wireworms. The same wireworm damage assessment was used for both sets of trials. The assessment involved counting all the erosions/scars caused by wireworm feeding activity. The prevalent wireworm species studied was *Agriotes sordidus*. Our research is the first to demonstrate that *Crotalaria* can significantly reduce wireworm damage when cultivated as a cover crop preceding potato. It, therefore, represents an effective standalone practice, one that can also be used with complementary practices to produce potatoes with low wireworm damage without using synthetic insecticides, thus avoiding an undesirable impact on the environment and human beings. The inclusion of *Crotalaria* in the rotation gives additional positive agronomic and environmental effects in a period when few suitable cover crops are available.

## 1. Introduction

Wireworms (Coleoptera: Elateridae) are important polyphagous soil pests of a considerable number of arable crops in Europe and North America [1]. Europe’s most harmful species are in the genus *Agriotes* Eschscholtz, 1829: *Agriotes brevis* Candèze, *A. lineatus* L., *A. litigiosus* Rossi, *A. obscurus* L., *A. proximus* L., *A. rufipalpis* Brullé, *A. sordidus* Illiger, *A. sputator* L., and *A. ustulatus* Schäller [2]. *A. lineatus*, *A. obscurus,* and *A. sputator* are important in North America, as well [3].

Obviously, root and tuber vegetable crops (i.e., carrot, parsnip, potato, radish, sweet potato, turnip) are particularly susceptible to wireworm attacks. Potato tubers (*Solanum tuberosum* L.) can be damaged by tunnelling after wireworm attacks, causing a major loss of their market value (based on current standards) due to lots being downgraded, resulting in substantial economic loss for growers [4].

Potato susceptibility to wireworm attacks depends on the cultivar [5]. Research on wireworm biology and ecology behavior plays a key role in developing Integrated Pest Management (IPM) packages [6].

For many years, the phytosanitary strategies for protecting potato crops from wireworm damage were based solely on synthetic insecticides [7] due to their long-term persistence in the soil. However, they are also massively detrimental to soil biodiversity and the environment [8,9]. In addition, since an EU-wide ban on Fipronil came into effect in 2014, extending to Thiamethoxam and Ethoprophos in 2018–2019 and to Chlorpyrifos in 2020, potato growers have complained of increased damage at harvesting [10].

The European Union made the implementation of IPM principles (Directive 2009/128/CE) compulsory in 2014, but today, IPM (i.e., prevention, monitoring, and replacement of synthetic insecticides) is still not being correctly adopted in the field. This failure is probably the main cause of current pest damage by wireworms, which is no longer dealt with by persistent broad-spectrum insecticides [11].

Agronomic preventive strategies should first be put in place to reduce wireworm damage risk to crops and, thus, the need for further crop protection. However, despite the availability of IPM packages based on agronomic scientific innovation, which can reduce wireworm risk, they are still little implemented on ordinary farms [6]. These packages include the use of bioactive cover crops that may keep pest populations (e.g., wireworms [12] and nematodes [13]) below damage thresholds and simultaneously supply other important agronomic benefits. Legume species and crops within crop rotation are crucial to maintaining or developing a more sustainable and complex agroecosystem. This is not only because they can host the microbial symbiosis that can biologically fix nitrogen at root level and thus reduce the use of synthetic nitrogen [14], but also because they can promote rhizospheric biological and enzymatic activities via the modification of rhizosphere pH (acidification) and the phosphorous pool, leading to a range of impacts [15].

Sunn hemp (*Crotalaria juncea* L.) is a tropical Asian plant of the Fabaceae legume family, also known as Indian hemp, Madras hemp, or brown hemp. *Crotalaria juncea* is native to India, where it is farmed as a source of fiber, which is obtained from its stem [16]. India is the main cultivator, but it is also farmed in neighboring South Asian countries. Despite being found in hepatic phytotoxins, mainly 1,2-dehydropyrrolizidine esters from pyrrolizidine alkaloids (PAs) [17], *Crotalaria* spp. is used either alone or combined with other plants in traditional herbal remedies in Asia and Africa [18,19]. *Crotalaria* is also increasingly used as green manure due to its high nitrogen content, thus improving soil quality and crop-system sustainability, as it reduces nitrogen inputs, N_2_O emissions, and nutrient leaching [20]. Furthermore, Arone et al. (2024) [21] recently found that *Crotalaria* green manure can improve the soil microbial community, increasing the genera involved as degraders of organic matter, as well as promoting plant growth and biological control of pathogens. *Crotalaria* spp., especially *C. juncea*, was one of the pyrrolizidine alkaloid plants that had already been proven to reduce other phytophagous pests, such as nematodes [22,23,24]. However, some studies demonstrated that when *Crotalaria* was added to the crop rotation in fields infested with the plant nematode, positive effects were noted on yields [25], but nematode populations were unaffected [25]. Interestingly, specialized herbivore insects, such as the larvae of the arctiid moth *Utetheisa ornatrix* L. (Lepidoptera: Erebidae), sequester pyrrolizidine alkaloid compounds from their *Crotalaria* (Fabaceae) host plant and retain them through metamorphosis into the adult stage. Both adult males and females provide PAs for egg protection [26], an example of the highly complex interactions between this plant and insect fauna.

This paper intends to demonstrate that *Crotalaria juncea* can significantly reduce the risk of wireworm damage for subsequent potato or other susceptible crops, and that it can thus represent a further agronomic prevention strategy to prevent wireworm potato damage.

## 2. Materials and Methods

### 2.1. Crotalaria juncea Characteristics—Sampling and Content Analysis

*Crotalaria juncea* plant samples for content analysis were recovered from the experimental fields where *Crotalaria* was planted before the potato crop. Part of the *Crotalaria* biomass samples was used for the pot trials.

#### 2.1.1. *Crotalaria juncea* Plant Sampling

On 14 October 2018, four *Crotalaria* plants were randomly collected from a 50 cm-long segment of a row in the trial field in Dugliolo di Budrio (Bologna province, Italy). They were gently uprooted at flowering time (60–67 BBCH) before the standing plants were chopped and incorporated into the soil as green manure. For each plant, (i) leaves, (ii) stems, (iii) roots, and (iv) flowers were separated and oven-dried at 40 °C until their weight was constant in order to determine biomass production expressed as dry weight. The samples were then finely ground to 0.5 mm, and a sub-sample was collected to determine nitrogen (N) content by dry combustion with a TruSpec^®^ Micro CHN elemental analyzer (LECO Corporation, St. Joseph, MI, USA). The samples were sent to the National Reference Laboratory for Plant Toxins in Food, Food Chemical Department, at Bologna’s Istituto Zooprofilattico Sperimentale della Lombardia e dell’Emilia-Romagna (IZSLER).

#### 2.1.2. Sample Preparation

An aliquot of leaves, stems, roots, and flowers from each plant was carefully homogenized and weighed (1 ± 0.1 g). Each sample was extracted with 25 mL of 0.2% formic acid in water, vortex-mixed, shaken for 45 min, and ultracentrifuged at 20,000 rpm at room temperature for 10 min. The extracts were then loaded onto SPE columns. A 10 mL aliquot of each sample was transferred and passed through Oasis MCX solid-phase extraction cartridges (Mixed-Mode Cation exchange sorbent, 60 mg/3 mL, Waters) that were preconditioned with 3 mL methanol and activated with 3 mL of 0.2% formic acid. An aliquot of the extract was then loaded into each cartridge. The cartridges were washed with 3 mL water and 3 mL methanol and eluted with 2.5 mL of methanol containing 2.5% ammonia. The eluates were evaporated to dryness under a gentle flow of nitrogen in a water bath at 40 °C, then dissolved in 1 mL of a water/methanol (95:5 *v*/*v*) mixture and transferred into vials for LC-MS/MS analysis. A quality-control sample, consisting of a spiked sample at the limit of quantification (LOQ) of 3 µg/kg for dried herbs, was assessed with each batch analysis.

#### 2.1.3. Analytical Standards and Reagents

Analytical standards of 42 pyrrolizidine alkaloids (PAs) and their N-oxides were obtained from Phytolab (Vestenbergsgreuth, Germany): Echimidine (Em), Echimidine-N-oxide (Em-ox), Echinatine (En), Echinatine-N-oxide (En-ox), Europine (Eu), Europine-N-oxide (Eu-ox), Heliosupine (Hs), Heliosupine-N-oxide (Hs-ox), Heliotrine (Ht), Heliotrine-N-oxide (Ht-ox), Indicine, Indicine-N-oxide, Integerrimine (Ir), Integerrimine-N-oxide (Ir-ox), Intermedine (Im), Intermedine-N-oxide (Im-ox), Lasiocarpine (Lc), Lasiocarpine-N-oxide (Lc-ox), Lycopsamine (Ly), Lycopsamine-N-oxide (Ly-ox), Retrorsine (Rt), Retrorsine-N-oxide (Rt-ox), Rinderine (Ri), Rinderine-N-oxide (Ri-ox), Senecionine (Sn), Senecionine-N-oxide (Sn-ox), Seneciphylline (Sp), Seneciphylline-N-oxide (Sp-ox), Senecivernine (Sv), Senecivernine-N-oxide (Sv-ox), Senkirkine (Sk), Spartioidine (Sp), Spartioidine-N-oxide (Sp-ox), Usaramine (Us), Usaramine-N-oxide (Us-ox), Jacobine (Jb), Jacobine-N-oxide (Jb-ox), Trichodesmine (Td), Erucifoline (Er), Erucifoline-N-oxide (ErNO), Monocrotaline (Mc), and Monocrotaline-N-oxide (McNO).

Methanol (LC-MS grade) was purchased from VWR Chemicals (Rosny-sous-Bois-cedex, France); formic acid (99%) and ammonia (30%) from Carlo Erba Reagents (Val de Reuil Cedex, France); and ammonium formate (analytical grade) from Sigma-Aldrich (St. Louis, MO, USA). Ultrapure water was purchased from an EvoQua Water Technologies system (Diessechem, Milan, Italy).

Stock standard solutions for each PA and its N-oxide derivatives (1000 µg/mL) were prepared in methanol. Working standard solutions, containing a mixture of PAs at varying concentrations, were prepared in a water/methanol (95:5 *v*/*v*) mixture. All solutions were stored at −20 °C.

#### 2.1.4. Performance Evaluation

The LC-MS/MS method was evaluated for specificity, recovery rates, linearity, repeatability, laboratory reproducibility, and limit of quantification (LOQ), in accordance with the EURL-MP guidance document on performance criteria for plant toxins, and Commission Regulation (EU) 2023/915 [27,28].

#### 2.1.5. LC-MS/MS System and Chromatographic Conditions for Analysis

Analysis was conducted using an Acquity I-Class ultra-performance liquid chromatograph (UPLC) coupled with an Xevo TQXS mass spectrometer (Waters Corporation, Milford, MA, USA). Chromatographic separation was performed on an Acquity UPLC C8 100 mm × 2.1 mm, 1.7 µm column (Waters Corporation, Milford, MA, USA). Data acquisition and processing were carried out using TargetLynx software (version 4.2). The mobile phase consisted of 5 mM ammonium formate with 0.1% formic acid in water (A) and 5 mM ammonium formate with 0.1% formic acid in methanol (B). The total runtime was 20 min. The flow rate was set to 0.3 mL/min, and the injection volume was 1 μL. The ESI source was operated in positive ionization mode with the following instrumental parameters: capillary voltage of 1.0 kV, cone voltage of 20 V, source temperature of 150 °C, and desolvation temperature of 600 °C.

### 2.2. Pot Experiments

Two pot trials were conducted in semi-controlled conditions. Trial 1 was conducted between 16 July 2013 and 18 December 2013 in a laboratory at the Sasse Rami Experimental Farm, Ceregnano (Rovigo province), Italy (coordinates 45.048487, 11.880473). Trial 2 was conducted between 27 October 2020 and 12 December 2020 in a laboratory in Adria (Rovigo province), Italy (coordinates 45.059963, 12.050230). Each pot hosted one treatment. The pots were located in a PVC-covered tunnel until 19 November; after that, they were stored inside a building. The larvae used for the experiment were produced in rearing cages over the previous two years with the method described by Furlan 2004 [29]. The experiment was set up as in Civolani et al. 2023 [30] using a randomized block design with four blocks, in which each one was a homogeneous replication that included all the treatments. Twelve pots were used for each trial. The substrate used was a mixture of 70% loam collected in untreated local fields and 30% river sand kept at maximum water capacity. The containers were 11.5 cm-high plastic pots with a 12 cm top diameter, 10 cm bottom diameter, and 1.1 L volume; holes in the pots’ bases were plugged with cotton tissue, which prevented larvae from escaping and allowed excess water to drain.

#### 2.2.1. Trial 1—Effect of Growing *Crotalaria* Plants on Wireworms

Treatments:*Crotalaria juncea*: 10 seeds per pot with no larvae (wireworms);*Crotalaria juncea* + larvae at sowing: 10 seeds per pot with six *Agriotes litigiosus* Rossi larvae introduced two days after pot preparation. Larvae head width ranged from 0.90 mm to 1.20 mm;*Crotalaria juncea* + larvae after sowing: plants thinned out at three per pot at the 6–7 true leaf stage, immediately followed by introduction of 12 *A. litigiosus* Rossi larvae from the same group used for Treatment 2 introduced.

The temperature pattern during the experiment is described in Figure 1.

#### 2.2.2. Trial 2—Effect of Incorporating *Crotalaria* Biomass on Wireworms

The biomass was collected from freshly chopped *Crotalaria juncea* grown in the cultivated fields in Budrio, Bologna province, Italy, in October (Table 1). It included all parts of the plant. The biomass was immediately stored in refrigerated containers for transportation and then in a stable refrigerator until it was time to incorporate it into the pots. Still frozen, it was broken by hand and mixed evenly with the entire volume of pot soil. The experiment temperature ranged from 5 °C to 30 °C.

A 3–4 cm layer of soil was laid in the bottom of 12 pots (3 treatments × 4 replications), a Monalisa cultivar tuber was placed in each one, and the pot was filled with another 3–4 cm of soil. The entire tuber surface was covered with soil in each pot. All tubers used were untreated A-class certified tuber seed, carefully selected to ensure that each genotype comprised similar shapes and sizes (about 30–35 mm); one tuber was used per pot.

Treatments:(1)*Crotalaria juncea* Madras^®^ (Nutrien Italia, Livorno, Italy) (CROTALARIA), 9.25 g per liter of soil (i.e., 37 t/ha fresh weight or 10.175 g/pot);(2)*Metarhizium brunneum* strain Ma43 BIPESCO 5 powder (Agrifutur, Alfianello, Italy) (METARHIZIUM), 0.2 g/m^2^ at seeding and 0.1 g/m^2^ once every 3 weeks, added to 150 mL water;(3)Untreated tubers.

Number of larvae per pot: six *Agriotes sordidus* larvae in the 7th–8th instar [29].

#### 2.2.3. Pot Inspections and Surveys

##### Larvae Survival/Mortality

At the end of both Trials 1 and 2, the pot contents were turned out onto a towel, the soil removed by hand, and the larvae were identified and divided into three groups:Alive and moving (left on the towel, they moved away quickly);Dying (left on the towel for a minute, they could not move in a specific direction), or almost immobile but alive;Dead.

Missing larvae were calculated by the difference.

##### Trial 1—Survey on *C. juncea* Plants

The vegetative parameters of each *Crotalaria juncea* plant (height, root length, shoots fresh weight, root fresh weight) and larval plant damage (root and collar erosions) were assessed per pot.

##### Trial 2—Survey on Potato Tubers

The tubers were inspected every week; any buds that could condition larval feeding were removed, and rotten tubers were replaced with new ones when appropriate. At the end of the experiment, the tubers were removed from their pots for final evaluation of wireworm erosion. Parameters were collected for each potato tuber, including:-Number of superficial scars/holes;-Number of deep scars/holes.

Each scar/hole was categorized as described in Table 2.

When an erosion was superficial (1–3 mm deep), it was considered “small”, whatever its width. The sum of ordinary and large erosions was considered “severe damage/erosion”.

### 2.3. Field Experiments

#### 2.3.1. *Crotalaria* Cultivation Information

*Crotalaria juncea* Madras^®^ was sown in 50% of the plots after winter cereals. The details of *Crotalaria* cultivation in 2018 and in 2019 are given in Table 1. The seed dosage was 25 kg ha^−1^ in both seasons. *Crotalaria* biomass was chopped with a rear shredder machine; mulch was air-dried for 48 h and then incorporated into the soil by ploughing 30–35 cm deep.

#### 2.3.2. Potato Cultivation and Experiment Information

The experiments were carried out in 2019 and 2020 in one of Italy’s most important potato-producing areas. The farm was located in Dugliolo di Budrio (Bologna province), Italy. Its main characteristics are described in Table 1. Each plot was at least 240 m^2^ (plot length 27 m per plot width 9 m). The experimental field/trial plots were laid out in randomized blocks with six replications. The Universa cultivar (Bretagne Plants Innovation, Hanvec, France) was chosen as a representative cultivar of the area, given its high yield. In fact, it is grown across the Italian peninsula, with high yields in all environments, and is one of the most widely used cultivars by Italian farmers. It is also a mid-early cultivar and, thus, is particularly susceptible to wireworm damage. After the potatoes were planted, row-ridging was carried out. Seed spacing was 90 cm between rows and 34 cm between plants; tuber seeds were uncut A-class certified 45–55 mm Ø. Local agronomic practices were used on all experimental field/trial plots homogeneously at each site.

The preceding crops on a two-year basis were sugar beet, followed by durum wheat, and sweet corn, followed by durum wheat, in 2019 and 2020, respectively. Green manure (Table 1) was then sown on part of the experimental plot, and potato fields were set up the following year. Sowing dates were 27 February and 3 March in 2019 and 2020, respectively. At planting time, 140 kg N, 185 kg P_2_O_5_, 199 kg K_2_O, and 28 kg MgO per hectare were distributed. At row-ridging, a further 60 kg N, 100 kg K_2_O, and 33 kg MgO were applied. At the same time, a mixture of synthetic chemical herbicides (Pendimethalin + Aclonifen + Metazachlor) was distributed in order to control weeds.

The various treatments are reported in Table 3. Biofence FL treatment was applied using a drip irrigation system.

#### 2.3.3. Assessment of Wireworm Species/Level

Bait traps were deployed to assess wireworm populations in spring or autumn before planting. A total of thirty-six traps were buried in each field-trial area in October 2018 as per Furlan [31].

#### 2.3.4. Estimation of Soil Pest Damage to Potato Tubers

See description of the method used in Pot Inspections and Surveys Trial 2—Survey on Potato Tubers (Section 2.2.3).

### 2.4. Larvae Identification

All the larvae used in the pot trials and found in the field trials were identified with a specific key [32].

### 2.5. Statistical Analysis

Data management and initial analysis were performed using Microsoft Excel [33]. Data normality was assessed using the Shapiro-Wilk test [34], and homogeneity of variances was evaluated with Levene’s test before proceeding with further analysis. Since the data did not follow a Gaussian distribution and showed heterogeneity of variances, ANOVA was performed on rank-transformed values [35,36]. Rank means were then separated using Tukey’s HSD test (*p* < 0.05). All data were processed in R [37]. In Table 8, the Median Absolute Deviation (MAD) was used instead of the standard deviation to better describe variability in the presence of a skewed data distribution. Data-processing was performed with STATGRAPHICS 19. 

## 3. Results

### 3.1. Pot Trials

#### 3.1.1. Trial 1

Table 4 presents the effects of wireworms on various growth parameters of *Crotalaria juncea* plants. Wireworms were able to feed on the *Crotalaria* plants, but caused damage that did not prevent the plants from growing (Table 3). The larvae feeding on *Crotalaria* plants could molt and grow, with the mortality rate being low (Table 4).

Table 5 presents data on wireworm survival and development across the two treatments where wireworms were introduced, either at sowing or after plant emergence. Examining the number of live wireworms retrieved at the end of the experiment, no significant differences were observed between the two treatments, with median values of 6.0 and 8.5 for wireworms introduced at sowing and after emergence, respectively. Similarly, the number of dead wireworms and adult wireworms did not differ significantly between treatments. When considering the total number of wireworms retrieved (alive, dead, and adult), the median values were 4.0 for the treatment with wireworms introduced at sowing and 9.0 for the treatment with wireworms introduced after emergence. However, these differences were not statistically significant. The treatment with wireworms introduced after emergence had a significantly lower median number of exuviae (0.5) compared to the treatment with wireworms introduced at sowing (1.0). Despite the difference in exuviae, the percentage of alive wireworms did not differ significantly between treatments, with median values of 66.67% for wireworms introduced at sowing and 70.83% for wireworms introduced after emergence.

#### 3.1.2. Trial 2

Trial 2 evaluated the effects of incorporating chopped *Crotalaria juncea* plant tissues into potting soil on wireworm damage to potato tubers and larval survival. Three treatments were compared: (1) *Crotalaria* incorporation; (2) Application of the bioinsecticide *Metarhizium brunneum*; and (3) Untreated control.

The results presented in Table 6 clearly demonstrate the efficacy of the *Crotalaria* treatment in reducing tuber damage caused by wireworm feeding and erosion. The median number of total erosions per tuber was significantly lower (up to four times) in the *Crotalaria* treatment than in the untreated control. Similarly, the number of severe erosions (ordinary + large) was also significantly reduced by the *Crotalaria* treatment. The *Metarhizium* treatment showed intermediate values for the erosion parameters, which were lower than the untreated control but higher than the *Crotalaria* treatment.

This suggests that while *Metarhizium* may provide some level of protection against wireworm damage, the *Crotalaria* treatment was more effective overall in preventing tuber erosion.

Table 7 presents data on wireworm larval survival across the treatments. There were no statistically significant differences in the number of alive, dying, dead, or missing larvae between the treatments.

### 3.2. Field Experiments

#### 3.2.1. *Crotalaria* Characteristics: Biomass Production and Alkaloid Analysis Results

Table 8 shows the quantification of pyrrolizidine alkaloids (PAs) in various organs of *Crotalaria juncea*. It reveals a highly skewed distribution of the data. This is evidenced by the large discrepancies between the median and mean values, indicating a non-normal distribution. To better describe the variability under these conditions, the Median Absolute Deviation (MAD) was used instead of standard deviation.

Although the median concentration of most alkaloids in the roots was zero, mean values were substantially higher, as observed with compounds such as Lycopsamine (mean: 40.18 µg kg^−1^, median: 0.0) and Trichodesmine (mean: 1603.28 µg kg^−1^, median: 291.95). This pattern reflects the presence of a limited number of high outliers in an otherwise low-concentration background, a trend consistent across most compounds and tissues (Table 8).

Trichodesmine was the predominant alkaloid, with particularly high concentrations in roots (median: 291.95 µg kg^−1^), decreasing progressively in stems (50.25 µg kg^−1^), leaves (9.40 µg kg^−1^), and becoming nearly absent in flowers (median: 0.0 µg kg^−1^). This compound is the main contributor to the total alkaloid content across all plant parts, as already reported in the literature [38].

The total alkaloid concentration in the whole plant followed a similar pattern, with median values ranging from 808.75 µg kg^−1^ in roots to 10.45 µg kg^−1^ in flowers. On a per-hectare basis, root biomass contributed the most to total alkaloid yield (0.75 g ha^−1^), while flower contribution was negligible (0.0 g ha^−1^ median).

Despite the generally low median values of many alkaloids—often zero—the relatively low MAD values indicate that most observations are clustered around these low values. Therefore, the sporadic occurrence of high concentrations does not represent a general pattern but, rather, rare events. This variability can be attributed to the Madras^®^ genotype used and the fact that its population characteristics are not yet stable. The term “cultivar” cannot be used here, since this plant retains typical characteristics of wild species, particularly no good genetic stability in its phenological and chemical traits.

**Table 8 insects-16-00674-t008:** Concentration of pyrrolizidine alkaloids (µg kg^−1^ DM) in different organs (root, stems, leaves, flowers) of *Crotalaria juncea*. For each compound, the median is reported outside the parentheses and the mean inside. Variability is expressed as Median Absolute Deviation (MAD). Total alkaloid content is also reported as g ha^−1^ (grams per hectare) for each plant part.

		Roots	MAD	Stems	MAD	Leaves	MAD	Flowers	MAD
Intermedine	(µg kg^−1^)	0.0 (10.25)	0.00						
Intermedine N-oxide	(µg kg^−1^)	2.0 (40.43)	2.00						
Lycopsamine	(µg kg^−1^)	0.0 (40.18)	0.00						
Lycopsamine N-oxide	(µg kg^−1^)	0.0 (44.58)	0.00						
Monocrotaline N-oxide	(µg kg^−1^)	0.0 (4.95)	0.00	0.0 (2.33)	0.00				
Isatidine (Retrorsine N-oxide)	(µg kg^−1^)			0.0 (2.80)	0.00	0.0 (3.73)	0.00	6.45 (12.40)	3.85
Seneciphylline	(µg kg^−1^)			0.0 (3.30)	0.00	0.0 (11.30)	0.00	0.0 (16.18)	0.00
Senecionine N-oxide	(µg kg^−1^)	0.0 (34.45)	0.00						
Senecivernine	(µg kg^−1^)	0.0 (50.45)	0.00						
Trichodesmine	(µg kg^−1^)	291.95 (1603.28)	46.05	50.25 (360.93)	12.25	09.40 (47.68)	1.70	0.0 (1.38)	0.00
Total Alkaloids in whole plant	(µg kg^−1^)	808.75 (2053.83)	512.80	74.65 (377.78)	36.65	69.50 (77.73)	60.10	10.45 (55.93)	2.75
Total Alkaloids per hectare	(g ha^−1^)	0.75 (2.97)	0.36	0.56 (3.17)	0.33	0.23 (0.34)	0.20	0.0 (0.02)	0.00

*Crotalaria* has shown particularly high biomass production under the growing conditions of the Po Valley, with total fresh matter exceeding 50 Mg ha^−1^ and dry matter reaching 14.4 Mg ha^−1^, as shown in Table 9. Total fresh biomass production reached a remarkable 54.6 ± 9.9 Mg ha^−1^ (Table 9). When converted to dry weight, total biomass production was 14.4 ± 3.4 Mg ha^−1^.

Examining the individual plant organs, it is evident that the stems contributed the largest share to total fresh biomass, with 29.6 ± 7.1 Mg ha^−1^, followed by leaves (18.4 ± 2.7 Mg ha^−1^), roots (4.5 ± 0.2 Mg ha^−1^), and flowers (2.1 ± 0.6 Mg ha^−1^). A similar pattern was observed for dry weight, with stems accounting for 7.7 ± 2.1 Mg ha^−1^, leaves 4.9 ± 1.1 Mg ha^−1^, roots 1.4 ± 0.2 Mg ha^−1^, and flowers 0.5 ± 0.2 Mg ha^−1^.

In terms of nitrogen content, the leaves contributed the highest amount (191.5 ± 42.5 kg ha^−1^), followed by stems (123.6 ± 34.1 kg ha^−1^), roots (12.2 ± 1.0 kg ha^−1^), and flowers (22.5 ± 7.5 kg ha^−1^). Total nitrogen contribution from the entire *Crotalaria juncea* crop was an impressive 349.9 ± 82.7 kg ha^−1^.

The flowers and leaves had the highest proportion of nitrogen (4.50% and 3.91%, respectively). As a result, these organs provided significant nitrogen input when incorporated into the soil, despite their relatively lower biomass compared to other plant parts. Although the stems had a lower proportion of nitrogen (1.61%), they contributed substantially to overall nitrogen input due to their high biomass production of 7.7 Mg ha^−1^ (dry weight). The roots had the lowest proportion of nitrogen (0.87%) but still played a role in nitrogen contribution to the soil due to their considerable biomass of 1.4 Mg ha^−1^ (dry weight).

#### 3.2.2. Assessment of Wireworm Species/Density

All the wireworms caught by bait traps or found in damaged tubers at harvest in the studied fields belonged to *Agriotes sordidus* Illiger (Table 10).

#### 3.2.3. Effect of *Crotalaria* Incorporation on Potato Damage by Wireworms

The data in Table 11 and Table 12 clearly demonstrate that plots where *Crotalaria juncea* was cultivated prior to potato planting had the lowest median of severely damaged tubers, both when the insecticide Mocap^®^ was applied and not applied. Up to half of the damage occurred in untreated plots. The untreated plots, none of which had any green manure incorporated, exhibited relatively high levels of severe wireworm damage (up to 85.7% without insecticide).

The application of Mocap® appeared to enhance the protective effect of *Crotalaria juncea*, leading to the lowest incidence of severely damaged tubers among all treatments: 10.43% in Trial 1 (Table 11) and 14.41% in Trial 2 (Table 12).

The average values provided in parentheses exhibit more variability compared to the median values, which are less influenced by extreme data points. This discrepancy highlights the importance of using appropriate statistical methods, such as rank transformation, to account for potential deviations from normality and heterogeneity of variances in the data.

The median values in Table 13 and Table 14 illustrate a clear distinction between the untreated control plots and those where *Crotalaria juncea* was incorporated. The untreated plots exhibited a higher number of severe erosions per tuber, with a median of 3.48 (no insecticide, no Biofence FL), 2.64 (Biofence FL), 4.17 (insecticide, no Biofence FL), and 2.00 (insecticide + Biofence FL). The overall median for the untreated control was 2.79 severe erosions per tuber.

In contrast, the plots where *Crotalaria juncea* was incorporated as a green manure showed a significantly lower number of severe erosions per tuber, with median values ranging from 1.79 (insecticide + Biofence FL) to 2.56 (insecticide, no Biofence FL). The overall median for the *Crotalaria* treatment was 1.96 severe erosions per tuber, which was significantly lower than the untreated control, as indicated by the different letters assigned to the median values based on the statistical analysis.

## 4. Discussion

*Crotalaria juncea* significantly reduced tuber damage by wireworms, both when incorporated as chopped plant pieces in pots (Table 6) and when planted the previous summer at field level (Table 11, Table 12, Table 13 and Table 14), but no wireworm mortality was observed (Table 7).

The results in Table 14 clearly demonstrate that the incorporation of *Crotalaria juncea* as a green manure prior to potato cultivation significantly reduces the number of severe wireworm erosions per tuber, corroborating the findings from Table 13 and Field Trial 1 (Table 11 and Table 12).

This protective effect was observed across all treatment combinations, further strengthening the evidence for the potential of *Crotalaria juncea* to be used as an effective agronomic practice for reducing wireworm damage in potato crops.

These clear effects occurred despite the high variability of pyrrolizidine alkaloids (Table 8 and Table 9). The growing *Crotalaria* plants allowed wireworms to grow and molt, without causing conspicuous larval mortality (Table 4 and Table 5). *Crotalaria* as a growing cover crop and incorporated plants did not significantly affect wireworm populations, suggesting that the observed differences in damage are likely due to plant resistance mechanisms rather than to direct effects on wireworms. In any case, the pyrrolizidine alkaloids, which likely play a role in this indirect effect, need to stay effective in the soil for many months.

Furthermore, while *Agriotes lineatus* larvae did not appear to experience significant direct adverse effects, their feeding on roots of plants containing pyrrolizidine alkaloids led to a reduction in the total concentration of alkaloids in the epigeal portion of the plant and a change in their chemical composition, with potential ecological repercussions at the level of multitrophic interactions [39]. This implies that the feeding activity of wireworms during the *Crotalaria* growing period may impact the potential effects of *Crotalaria* incorporated residues on subsequent wireworm populations.

The content of pyrrolizidine alkaloids in the genus *Crotalaria* has been analyzed under different aspects and conditions. For example, the trichodesmine content in the plant pods was evaluated according to the cultivar and the sowing time in a recent study [40]. Furthermore, some studies investigated the concentrations of major dehydropyrrolizidine alkaloids in roots, stems, leaves, and seeds of *Crotalaria juncea* cv. ‘Tropic Sun’ [41]. However, it is still unclear how stable the accumulation of these secondary metabolites in plant tissues is in each vegetative phase; the same can be said regarding the extent to which stability varies between different plants of the same cultivar and between different accessions. Interestingly, the aforementioned study [41] found concentrations of trichodesmine, the most abundant alkaloid in our determinations, of 0.007%, 0.013%, and 0.001% in DM in roots, stems, and leaves, respectively (Table 8). These values seem to be comparable to our findings, despite the high variability of the determinations between different plants. The high difference in trichodesmine content between different plants of the same population could also explain the low mortality of larvae found in pot trials (Table 7). It is also possible that other biological mechanisms play a role in reducing both trophic activity by larvae and reproductive capacity in the presence of *Crotalaria*. Although the available literature mainly focuses on how pyrrolizidinic alkaloids interact with phytophagous insects that feed on the aerial parts of plants [39], some studies have already highlighted the efficacy of *Crotalaria* against soil-dwelling parasites such as nematodes [13,23]. *Crotalaria* is also a poor feeding source or non-host to a large group of other pests and pathogens, competes with weeds without becoming one, grows vigorously to provide good ground coverage for soil erosion control, fixes nitrogen, and is a green manure [15,21]. However, most *Crotalaria* species are susceptible to some nematode species, such as *Pratylenchus* spp., *Helicotylenchus* sp., *Scutellonema* sp., and *Criconemella* spp. [23].

As regards human health, isatidine (also known as retrorsine N-oxide) and trichodesmine are pyrrolizidine alkaloids found in some plants and are known to be toxic. In general, pyrrolizidine alkaloids, including retrorsine and, presumably, isatidine and trichodesmine, are known to cause liver damage. These compounds are metabolized in the liver to reactive metabolites that can bind to cellular macromolecules, causing hepatic necrosis, veno-occlusion, and, in some cases, carcinogenesis. No specific data are available regarding human intoxication cases directly attributable to isatidine, but it has shown potential carcinogenicity in rat tests. However, retrorsine, a related alkaloid, is known to be toxic to mammals [42]. As an example, pyrrolizidine alkaloids in *Senecio* genus plants, including isatidine, caused pulmonary oedema and liver damage. Literature has reported cases of intoxication by pyrrolizidine alkaloids after use of herbs containing alkaloids for medicinal purposes, and the consumption of cereals contaminated by toxic plants (*Crotalaria* spp., *Senecio* spp.). Significant poisoning epidemics were documented in India during the 1970s, due to contamination of millet by *Crotalaria nana*, and in Afghanistan between the 1970s and 1980s, due to contamination of wheat by *Heliotropium popovii* [43]. Recent studies investigated contamination of spinach crops in various Italian areas by other molecules belonging to the macro-class of alkaloids contained in the spontaneous weeds of the *Datura* genus [44]. Consequently, careful attention must be paid when introducing weeds and plants containing alkaloids as cover crops for bioactive use and to avoid potentially dangerous contamination for human health.

*Crotalaria juncea* is native to tropical and subtropical regions. It is characterized by a short-day photoperiodic flowering response in temperatures ranging from 15 °C to 27 °C. This means that it tends to flower and set seed only when day lengths are sufficiently short. At high latitudes, such as in Northern Italy, longer summer days can inhibit flowering and, consequently, the production of fertile seeds. A study conducted in the United States [45] reported that common cultivars of *Crotalaria juncea* do not reliably produce seeds at higher latitudes. *Crotalaria juncea* does not develop seed north of 28°N latitude [46]. Consequently, there is little threat that the plant will spread or become weedy within Italy, which extends from 36° N to 46° N.

Our study appears to confirm Kostenko et al. [39], who showed that common ragwort (*Jacobaea vulgaris* Gaertn.), another species containing PAs, had no significant effect on larval mortality after larvae fed on its roots. Our findings also suggest that active compounds remain in the soil long-term and somehow disturb larval feeding.

## 5. Conclusions

Our research demonstrated that *Crotalaria juncea* as a cover crop can significantly reduce tuber wireworm damage without affecting wireworm survival. This is the first reported demonstration of the potential role of *C. juncea* in reducing wireworm potato damage. A major role is likely played by the high pyrrolizidine-alkaloid content in *C. juncea* tissues, but this cause-and-effect relationship is far from being specifically proven. Future studies should therefore continue to investigate which compounds are involved in *C. juncea* bioactivity and what the mechanism of action is in controlled conditions.

To identify their “protection mechanisms”, isolated alkaloids should be put in contact (e.g., in vial trials) with wireworms and the impact on feeding behavior checked. Likewise, the effect of isolated pyrrolizidine alkaloids on plant tissues/tuber appetibility for wireworms should be studied. Although some contributions show that *Crotalaria* spp. cultivation promotes soil biodiversity and subsequent crop performance, probably due to the agronomic effects of cover crops enhanced by Leguminosae characteristics [14,15,21,25], its effect on soil microbiota of *Crotalaria* incorporation and cultivation should be further assessed.

In practice, *Crotalaria* represents an effective means to be used alone or with complementary plants [6] for producing potatoes with low wireworm damage without using synthetic insecticides, thus avoiding an undesirable impact on the environment and human health. In addition, the correct inclusion of *Crotalaria* in crop rotations has additional agronomical benefits, since it is one of the few summer Leguminosae available in Northern Italy.

## Figures and Tables

**Figure 1 insects-16-00674-f001:**
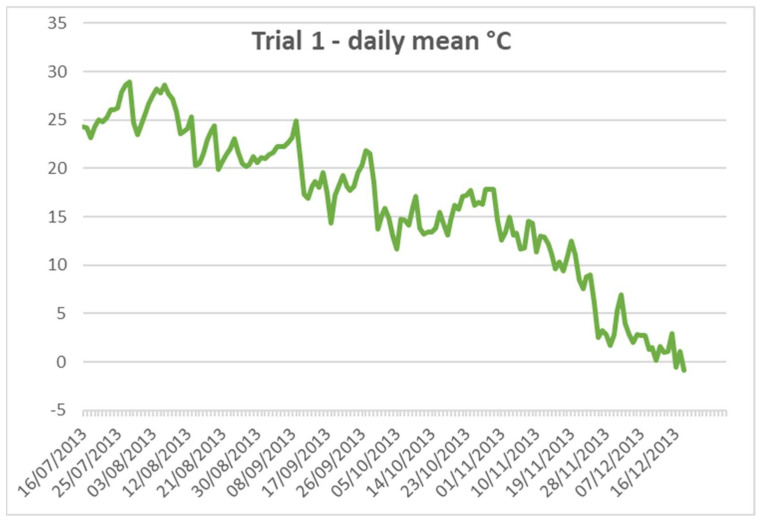
Temperature patterns during Pot Trials 1 (Year 2013) and 2 (Year 2020).

**Table 1 insects-16-00674-t001:** Agronomic information of *Crotalaria* cultivation in 2018 and 2019.

Geographical coordinates	44.37036 N–11.35094 E
Soil texture classification	sandy clay loam
Tillage operations prior sunn hemp sowing	Harrowing
Tillage operations after sunn hemp sowing	no operations
Preceding crops (Two-year basis)	2018, sugar beet→ durum wheat; 2019, sugar beet→ barley
Sunn hemp cultivar	Madras^®^
Seed spacing (cm)	4.5 × 22.5
Date of sowing	9 July 2018 27 June 2019
Mineral fertilization at sowing (kg/ha of N, P_2_O_5_, K_2_O)	36 (N) 36 (P_2_O_5_) 51 (K_2_O)
Mineral fertilization at plant height 25–30 cm (kg/ha of N, K_2_O, MgO)	70 (N)
Irrigation systems	rain gun hose reel
Disease chemical control (Active ingredient/commercial products/no. of foliar sprayings)	no treatments
Pest chemical control (Active ingredient/commercial products/no. of foliar sprayings)	no treatments
Date of plant chopping of green manure	21 October 2018 14 October 2019
Date of ploughing of green manure	23 October 2018 16 October 2019

**Table 2 insects-16-00674-t002:** Types of tuber erosion.

Type of Erosion	Diameter (mm)	Characteristics
Small	1–2	Open wound
Ordinary	2–5	Open wound
Large	>5	Open wound
Old	Variable	Healed, deformed hole due to early attack and tuber development

**Table 3 insects-16-00674-t003:** Treatments in field trials in 2019 and 2020. * Mocap^®^ granular insecticide (Certis, Saronno, Italy), *Ethoprophos* (15%), 35 kg/ha. ** Force Evo^®^ granular insecticide (Syngenta, Milan, Italy), *Tefluthrin* (0.5%), 15 kg/ha. *** Biofence FL (Nutrien Italia, Livorno, Italy), 20 L/ha in five applications per 4 L/ha, by drip irrigation. Liquid organic fertilizer based on Brassicaceae-derived biomasses that release Allyl-isothiocyanate. **** Madras^®^ (Nutrien Italia, Livorno, Italy), 25 kg/ha.

Field Code Treatment	Green Manure Prior to Potato Crop	Wireworm Control by		
		Application of Soil Insecticide in Furrow at Planting	Application of Soil Insecticide in Furrow at Row-Ridging	Drip Application
1-NT-NS	===	===	===	===
2-T-NS	===	Mocap^®^ *	Force Evo^®^ **	===
3-T-NS	===	Mocap^®^ *	Force Evo^®^ **	Organic Fertilizer *** (five times)
4-T-NS	===	===	===	Organic Fertilizer *** (five times)
1-NT-CJ	*Crotalaria* ****	===	===	===
2-T-CJ	*Crotalaria* ****	Mocap^®^ *	Force Evo^®^ **	===
3-T-CJ	*Crotalaria* ****	Mocap^®^ *	Force Evo^®^ **	Organic Fertilizer *** (five times)
4-T-CJ	*Crotalaria* ****	===	===	Organic Fertilizer *** (five times)

**Table 4 insects-16-00674-t004:** Pot trial. Effects of wireworm presence on plant growth parameters. Values represent medians (outside parentheses) and means (inside parentheses). Treatments were compared using ANOVA on rank-transformed data. Different letters indicate significant differences between treatments (*p* < 0.05, Tukey’s test). Lowercase and uppercase letters indicate significance at *p* < 0.05 and *p* < 0.01, respectively.

Treatment	Plant Height (cm)		Root Length (cm)		Damaged Plants (no.)		Damaged Plants (%)		Shoots Fresh Weight (g)		Roots Fresh Weight (g)		Total Fresh Weight (g)	
No wireworms	50.0 (48.63)	A	17.0 (18.83)	A	0.0 (0.0)	B	0.0 (0.0)	b	1.55 (1.59)	B	0.2 (0.25)	b	1.84 (1.84)	B
Wireworms at sowing	54.5 (52.3)	A	15.0 (17.16)	B	2.5 (2.25)	A	33.04 (30.8)	ab	1.78 (1.76)	B	0.21 (0.25)	b	1.95 (2.01)	B
Wireworms after emergence	48.0 (50.55)	B	14.0 (14.45)	C	2.0 (1.75)	B	66.67 (66.67)	a	2.05 (2.05)	A	0.23 (0.24)	a	2.33 (2.28)	A

**Table 5 insects-16-00674-t005:** Pot trial. Wireworm survival and development across treatments. Values represent medians (outside parentheses) and means (inside parentheses). Treatments were compared using ANOVA on rank-transformed data. Different letters indicate significant differences between treatments (*p* < 0.05, Tukey’s test). Lowercase and uppercase letters indicate significance at *p* < 0.05 and *p* < 0.01, respectively.

Treatment	Wireworms Introduced		Live Wireworms		Dead Wireworms		Adult Wireworms		Total Wireworms		Exuviae		Alive Wireworms (%)	
Wireworms at sowing	6.0 (6.0)		4.0 (3.25)	a	0.0 (0.0)	A	0.0 (0.25)	a	4.0 (3.5)	A	1.0 (0.75)	a	66.67 (58.33)	A
Wireworms after emergence	12.0 (12.0)		8.5 (8.25)	a	0.0 (0.25)	A	0.0 (0.0)	a	9.0 (8.5)	A	0.5 (0.5)	b	70.83 (68.75)	A

**Table 6 insects-16-00674-t006:** Pot trial. The table presents the median values of “Total erosions”, “Severe erosions”, and “Small erosions” for each tuber in each pot. Values represent medians (outside parentheses) and means (inside parentheses). Treatments were compared using ANOVA on rank-transformed data. Different letters indicate significant differences between treatments (*p* < 0.05, Tukey’s test). The letters next to the median values indicate groups of treatments that are not significantly different from each other, according to Tukey’s HSD test (*p* < 0.05). The original data were transformed into ranks before conducting the ANOVA. Treatments with the same letter belong to the same statistical group.

Treatment	Total Erosions		Severe Erosions		Small Erosions	
*Crotalaria*	2.00 (1.50)	b	0.50 (0.75)	b	0.50 (0.50)	B
Metarhizium	4.00 (3.50)	b	1.00 (1.25)	b	2.50 (2.25)	Ab
Untreated	9.50 (8.25)	a	4.00 (4.00)	a	4.00 (4.00)	A

**Table 7 insects-16-00674-t007:** Survival of the larvae in the pots. Values represent medians (outside parentheses) and means (inside parentheses). The probability level was assessed using ANOVA on ranked data. No significant statistical differences were found.

Treatment	Alive Larvae	Dying Larvae	Dead Larvae	Missing Larvae
*Crotalaria*	75.00 (70.83)	0.00 (4.17)	0.00 (0.00)	25.00 (25.00)
Metarhizium	58.33 (54.17)	0.00 (0.00)	0.00 (0.00)	41.67 (45.83)
Untreated	66.67 (62.50)	0.00 (0.00)	0.00 (4.17)	33.33 (33.33)

**Table 9 insects-16-00674-t009:** Biomass production and nitrogen content of *Crotalaria* by plant organ (roots, stems, leaves, and flowers), expressed in fresh weight, dry weight, and nitrogen contribution per hectare (mean ± standard deviation).

	UM	Roots	Stems	Leaves	Flowers	Total
Fresh weight	Mg ha^−1^	4.5 ± 0.2	29.6 ± 7.1	18.4 ± 2.7	2.1 ± 0.6	54.6 ± 9.9
Dry weight	Mg ha^−1^	1.4 ± 0.2	7.7 ± 2.1	4.9 ± 1.1	0.5 ± 0.2	14.4 ± 3.4
Nitrogen	kg ha^−1^	12.2 ± 1.0	123.6 ± 34.1	191.5 ± 42.5	22.5 ± 7.5	349.9 ± 82.7

**Table 10 insects-16-00674-t010:** Wireworms caught by bait traps or found in damaged tubers at harvest in the studied fields. * Median and mean inside parentheses. ** Variability is expressed as MAD (Median Absolute Deviation). n.a. = not assessed.

Year	Period	Trial	Wireworm Species		
			No. in Bait Trap *	MAD **	No. in Damaged Tubers
2018	Autumn	1	1.00 (0.81) *A. sordidus*	1.00	not applicable
2019	Harvest	1	n.a.	n.a.	4, *A. sordidus*
2020	Harvest	2	n.a.	n.a.	6, *A. sordidus*

**Table 11 insects-16-00674-t011:** Field Trial 1, 2019. Median percentage of tubers with severe wireworm damage (ordinary + large)/tuber in plots planted and not planted with *Crotalaria* when the insecticide Mocap^®^ was used and was not used. Corresponding mean values are in parentheses. The original data were transformed into ranks before conducting the ANOVA. Treatments with the same letter belong to the same statistical group. Uppercase letters indicate significance at *p* < 0.01.

	Mocap^®^					
Green Manure Prior to Potato Crop	YES		NO		Median (Average)	
Untreated	16.13 (15.59)		39.76 (38.84)		22.02 (27.22)	A
*Crotalaria juncea*	10.43 (12.11)		18.89 (21.28)		15.14 (16.69)	B
Median (average)	13.85 (14.60)	B	25.75 (30.06)	A		

**Table 12 insects-16-00674-t012:** Field Trial 2, 2020. Median number of wireworm severe erosions (ordinary + large)/tuber in plots planted and not planted with *Crotalaria* when the insecticide Mocap^®^ was used and was not used. Corresponding mean values are in parentheses. The original data were transformed into ranks before conducting the ANOVA. Treatments with no letter belong to the same statistical group. Lowercase and uppercase letters indicate significance at *p* < 0.05 and *p* < 0.01, respectively.

	Mocap^®^				
Green Manure Prior to Potato Crop	YES		NO		Median (Average)
NO	33.48 (14.41)	B	85.69 (85.33)	a	60.69 (60.01) a
*Crotalaria juncea*	14.41 (18.82)	B	59.15 (56.41)	a	31.73 (37.62) b
Median (average)	28.92 (26.75)	A	77.55 (70.87)	b	

**Table 13 insects-16-00674-t013:** Field Trial 2, 2020. Median percentage of tubers with severe wireworm damage (at least one ordinary or large erosion/hole). The treatments included BIOFENCE FL (applied via drip irrigation: NO/YES) and a combined chemical treatment with Mocap^®^ + Force Evo^®^ (NO/YES). Corresponding mean values are in parentheses. The original data were transformed into ranks before conducting the ANOVA. The probability level was assessed using ANOVA on ranked data. The separation of rank means was performed using Tukey’s HSD test (*p* < 0.05). Medians with the same letter belong to the same statistical group.

Severe Damage (%)	BIOFENCE FL (Drip Irrigation)		Mocap^®^ + Force Evo^®^		Median (Average)
	NO	YES	NO	YES	
Untreated	83 (69)	57 (57)	84 (83)	32 (33)	74 (62) a
*Crotalaria*	27 (34)	49 (40)	59 (54)	13 (17)	30 (37) b

**Table 14 insects-16-00674-t014:** Field Trial 2, 2020. Number of wireworm severe erosions (ordinary + large)/tuber. The treatments included BIOFENCE FL (applied via drip irrigation: NO/YES) and a combined chemical treatment with Mocap^®^ + Force Evo^®^ (NO/YES). Corresponding mean values are in parentheses. The original data were transformed into ranks before conducting the ANOVA (*p* < 0.05). Treatments with no letter belong to the same statistical group.

Severe Damage (%)	BIOFENCE FL (Drip Irrigation)		Mocap^®^ + Force Evo^®^		Median (Average)
	NO	YES	NO	YES	
Untreated	3.48 (3.49)	2.64 (3.20)	4.17 (4.52)	2.00 (2.17)	2.79 (3.35) a
*Crotalaria*	1.83 (2.28)	2.06 (2.26)	2.56 (2.71)	1.79 (1.83)	1.96 (2.27) b

## Data Availability

The original contributions presented in this study are included in the article. Further inquiries can be directed to the corresponding author.

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
