# Peer review of "Effectiveness of Sunn Hemp (Crotalaria juncea L.) in Reducing Wireworm Damage in Potatoes"

_insects, 2025, doi:10.3390/insects16070674_

Round 1
Reviewer 1 Report
Comments and Suggestions for Authors
Wireworms are one the serious pests of many crops including potato. Here in this study, the authors tested the Effectiveness of Sunn Hemp (Crotalaria juncea L.) in Reducing Wireworm Damage in Potatoes. Field research often occurs in uncontrolled environments which make it difficult to obtain good data. Here authors conducted a pot and field study to assess the potential of Sunn Hemp plant in IPM. The study contains lots of data that can contribute the literature. However, there are some points that the authurs need to clarify.
Why did the authors use Fungal entomopathogen here? why did you chose entomopathogenic fungi and why did not you chose entomopathogenic nematode? Please clarify.
The authors used a fungal entomopathogen in pots but in field insecticides? This disrupts the integrity of the study. Please clarify
Wireworm populations within a field often consist of a mix of species of Agriotes as well as wireworms belonging to other genera. So did the authors survey the field if the other species are already present in the field? I can not see any information about this.
Also, how did the autthors obtain a reasonable estimate of numbers of the larvae in field?
Please check pdf file for detailed comments.

Reviewer 2 Report
Comments and Suggestions for Authors
I find this manuscript interesting.
However, I think authors should emphasize more in the Introduction the importance of Crotalaria and its invasive status in Italy.
Does it have any application in European agriculture? Has anyone studied this plant as a crop or cover crop in Italy or Europe? Authors cite only the Brazilian example.
Additionally, are there any data suggesting that Crotalaria is safe to be cultivated in Italy, considering that invasive species cause significant damage to the environment nowadays?
The summary and abstract are written well.
In 47, what do you mean by complementary ones? Clarify or remove
ln 231 – what was the pot size in this experiment?
ln 321- potato variety, or rather cultivar?
Table 9: Please explain in the legend what the number of plants served as the basis for this calculation.
How did the authors assess the nitrogen content of Crotalaria? I could not find it in the methods.
Tables 11 and 12, the use of insecticide should be marked in the table title, otherwise the term Ethoprophos is not clear to the reader.
Tables 13 and 14 – the same comment, explaining insecticides.
ln 660-661 – “Our findings also suggest that active compounds remain in the soil long-term and somehow disturb larval feeding”. -- Which of your results support this statement?
